# Antifungal In Vitro Activity of *Phoradendron* sp. Extracts on Fungal Isolates from Tomato Crop

**DOI:** 10.3390/plants12030672

**Published:** 2023-02-03

**Authors:** Alma Leticia Salas-Gómez, César Alejandro Espinoza Ahumada, Rocío Guadalupe Castillo Godina, Juan Alberto Ascacio-Valdés, Raúl Rodríguez-Herrera, Ma. Teresa de Jesús Segura Martínez, Efraín Neri Ramírez, Benigno Estrada Drouaillet, Eduardo Osorio-Hernández

**Affiliations:** 1Research and Postgraduate, Faculty of Engineering and Sciences, Autonomous University of Tamaulipas, University Center Adolfo Lopez Mateos. Cd., Victoria 87120, Tamaulipas, Mexico; 2El Mante Superior of Technological Institute, Km 6.7, Highway Mexico 85, Quintero 89930, Tamaulipas, Mexico; 3School of Chemistry, Autonomous University of Coahuila, Saltillo 25280, Coahuila, Mexico

**Keywords:** cedar, ethanolic extracts, mesquite, mistletoe, oak, polyphenols

## Abstract

Synthetic chemicals are mainly used for the control of fungal diseases in tomato, causing the phytopathogens to generate resistance to the chemical active ingredient, with a consequent risk to human health and the environment. The use of plant extracts is an option for the control of these diseases, which is why the main objective of this research was to study an alternative biocontrol strategy for the management of plant diseases caused by fungi through obtaining polyphenol extracts from mistletoe plants growing on three different tree species—mesquite (*Prosopis glandulosa*), cedar (*Cedrus*), and oak (*Quercus*), which contain flavones, anthocyanins, and luteolin. The overall chemical structure of the obtained plant extracts was investigated by RP-HPLC-ESI-MS liquid chromatography. The antifungal effect of these extracts was examined. The target phytopathogenic fungi were isolated from tomato plantations located in Altamira, Tamaulipas, Mexico. The microorganisms were characterized by classical and molecular methods and identified as *Alternaria alternata*, *Fusarium oxysporum*, *Fusarium* sp., and *Rhizoctonia solani*.

## 1. Introduction

Tomato crop is one of the agricultural commodities with the highest economic importance worldwide; in addition, it offers benefits to human health due to its high content of potassium and antioxidants such as ascorbic acid, vitamin A, lycopene, and tocopherols [1]. However, this crop is attacked by more than 200 pathogens that affect its yield and quality, the most important being *Alternaria solani,* causing early blight, and *Rhizoctonia solani*, *Fusarium oxysporum,* and *Fusarium solani* causing wilt. These pathogens may cause yield losses of more than 60% [2], whereby, they are mostly controlled through the use of synthetic chemicals [3], which may have some weaknesses such as being a risk to human health, the environment, and beneficial organisms. This is why ecologically safe alternatives are being explored to control these plant pathogens [4]. In this context, biological control is an alternative that has shown efficacy for the management of pathogens that affect tomato plants [5]. One strategy to reduce the spread of disease-causing fungi in vegetables is the use of plant extracts, which are characterized by the presence of secondary metabolites such as alkaloids, saponins, and terpenoids that have fungistatic activity and/or induce plant defenses [6]. Extracts of indigenous medicinal plants of Pakistan such as *Justacia adhatoda*, *Azadriachta indica*, *Foeniculum vulgare*, and *Mentha spicata* have been used and the presence of various functional groups such as alcohols, carboxylic acids, esters, alkanes, and alkenes were determined by Fourier transform infrared spectroscopy (FTIR), which suppressed the growth of *Alternaria alternata* [7]. The use of these extracts can be taken as an option for the control of fruit rot in an environmentally friendly manner. These compounds are used by plants as a defense against biotic and abiotic factors, and are grouped into flavonoids, phenols, terpenes, alkaloids, lectins, and polypeptides [8]. On the other hand, *Phoradendron* sp. is widely used in medicine as an anti-inflammatory, alternative treatment for cancer, among other uses, because *Phoradendron* sp. contains flavonoids, phenolic acids, antioxidant, fatty acids, and anthocyanins, making it a good candidate for the development of products for phytopathogen control [9,10]. *Phoradendron* sp. is hemiparasitic and survives due to the nutrients obtained from the host, therefore, for this reason, its phytochemical and mineral content depend on the host [11,12]. This plant is distributed throughout the world, with *Viscaseae* and *Loranthaceae*, two prominent families of this species [12]. Some of the species of Phoradendron sp. found in northern and central Mexico are *Phoradendron bollanum* and *Viscum album* subs [13], which have been rarely studied, and as only their phytochemical contents have been reported, their potential for the control of phytopathogens has been little tested, although morolic acid has been reported as the main component (47.54% of the total composition of the acetone extract) of this plant. In addition, the identification of 19 known compounds has been reported: β-sitosteryl and stigmasteryl linoleates, β-sitosterol, stigmasterol, triacontanol, squalene, α- and β-amyrin, lupeol, lupenone, betulin aldehyde, betulon aldehyde, oleanolic aldehyde, betulinic acid, betulonic acid, moronic acid, morolic acid, oleanolic acid, flavonoids acacetin, and acacetin 7-methyl ether. According to the phytochemicals present in this species, the objective of this study was to evaluate in vitro the antifungal activity of *Phoradendron* sp. extracts against *Alternaria solani*, *Fusarium oxysporum*, *Fusarium* sp., and *Rhizoctonia solani*.

## 2. Results

### 2.1. Phytochemicals Present in Plant Extracts

The *Phoradendron* sp. extracts obtained from three different hosts possessed a variety of compounds that differed from each other (Table 1), where hydroxycinnamic acids (HCAs) were the family of compounds observed in the three *Phoradendron* sp., in addition to anthocyanins, flavonols, and flavones (Table 1).

### 2.2. Isolation of Phytopathogens

The phytopathogenic fungal isolates were isolated from tomato tissue and soil samples, which were classified as *Fusarium oxysporum* (roots), *Fusarium* sp. (soil), *Alternaria alternata* (leaves), and *Rhizoctonia solani* (stems) based on their morphological features. The characteristics of *Alternaria alternata* during the first days of growth were light brown, changing to dark brown from the fifth day, and it took approximately 8 days to fill the 80 mm diameter Petri dish (Figure 1a). Under the compound microscope, brown conidia with chain formation, straight, transverse, and longitudinal septa were observed (Figure 1b). The colony of *Fusarium* sp. presented sparse mycelium of cream coloration (Figure 1c), the microscopic characteristics observed were macroconidia of approximately 50 µm in length, and robust and rounded at the ends with five septa (Figure 1d). *Fusarium oxysporum* showed abundant purplish mycelium (Figure 1e), with macroconidia approximately 37 µm long with hooked apex and the presence of three septa (Figure 1f). *Rhizoctonia solani* showed light brown and dark brown ringed growth (Figure 1g), the hyphae branched at right angles, and there was always a septum in the branching of the hyphae near the point of origin, with a slight constriction in the branching, and no conidia or conidiophores were observed (Figure 1h). The length of hyphae between two septa ranged from 67.6 to 149.8 μm (mean 109.5 μm).

The results of the sequence analysis of the isolates corresponded to *Alternaria alternata* with 99.5% identity, *Fusarium oxysporum* 98.99%, *Fusarium* sp. with 100%, and *Rhizoctonia solani* with 92% in the NCBI general database (Table 2).

### 2.3. Inhibition of Fungal Growth

In the experiments with extracts and phytopathogenic fungi isolated from tomato plants, significant differences (*p* ≤ 0.05) were found, with different statistical groups (Table 3). The mycelial growth of *Alternaria alternata* was inhibited with the extract of mistletoe grown on cedar (CME) at 4000 ppm (35.60%), at the same dose the extracts of mistletoe grown on oak (OME) and mistletoe grown on mesquite (MME) inhibited the growth of this phytopathogen by 24 and 22%, respectively (Table 3). The results obtained show that the mistletoe extract has the capacity to produce active substances (polyphenols) that inhibit the growth of *Alternaria alternata*. With respect to MME against *Fusarium oxysporum*, significant statistical differences were found, since at a concentration of 4000 ppm, its growth was inhibited by 52.32%. It is worth mentioning that as the concentration increased, the inhibition increased, showing a dose-dependent effect. OME and CME at 4000 ppm inhibited pathogen growth by 40.6% and 13.13%, respectively (Table 3). The mistletoe extract grown on mesquite at concentrations from 250 to 1000 ppm did not show statistical differences for *Fusarium* sp. inhibition as the values were between 3.51 and 3.94%; however, at 2000 ppm, it showed an inhibition of *Fusarium* sp. of 11.66% while at 4000 ppm, the inhibition was 44.91%. For the mistletoe extract grown on cedar and oak, the inhibition at 4000 ppm was 26.1 and 24.1%, respectively (Table 3). For *Rhizoctonia solani*, the inhibition by the extracts was very small since the inhibition percentages had values of 1.70% at 4000 ppm with CME, for OME and MME at the same concentration, the values were 1.33 and 13.05%, respectively (Table 3).

### 2.4. Number of Conidia

The results of the evaluation of the effect on the production of conidia with the extracts at the different doses are shown in Table 4, where it was observed that *Alternaria alternata* at the dose of 4000 ppm of MME presented the lowest number of conidia (0.2 × 10^6^ conidia*mL^−1^), while *F. oxysporum* with MME at 500 ppm presented lower values (42 × 10^6^ conidia*mL^−1^) with respect to the other two extracts evaluated. Finally, *Fusarium* sp. with MME at 2000 ppm presented a lower number of conidia (4.6 × 10^6^ conidia*mL^−1^) compared to the other extracts.

## 3. Discussion

Due to the demand for food free of pesticide residues, there is a need for research on alternatives that do not cause damage to the environment, but above all, that do not represent a health risk. One option is plant-based extracts, since they are safe to use and are an option to replace dangerous synthetic chemical pesticides, especially to control fungal diseases that occur in crops [14]. Consequently, there is growing interest in substituting chemical-synthetic products with products made from natural plant extracts, since the plants from which they are made are available worldwide at low cost, and they do not have harmful effects on the fruit quality and human health, which would benefit producers by reducing the production costs [15].

The presence of flavonoids in the extracts evaluated gives them their antioxidant and antibacterial capacity [6,16]. *Phoradendron* sp. contains flavones, a group that has been studied and used in different areas, as it is responsible for its antimicrobial activity; other compounds such as lutein were found in the *Phoradendron* sp. extract from oak and *Phoradendron* sp. extract from mesquite and was also reported in *V. album* from Romania by Trifunschi et al. [17]. In addition, studies conducted with extracts of *Phellinus gilvus*, *Phellinus rimosus*, and *Phellinus badius* by Ayala et al. [18] showed an inhibitory effect against *Alternaria alternata* and determined the content of phenolic compounds, which presented values of 30.58, 28, and 26.48 mg QE/g equivalent, respectively, which also had results of 50% in inhibiting the growth of this pathogen.

In the studies conducted by Garcia et al. [6] of the phytochemicals present in extracts of Mexican mistletoe, hydroxycinnamic acids (HCA) were the compounds most present in the extract, and these compounds have also been reported in extracts of some mistletoes in previous research [19], to which immunomodulatory, antibacterial, hypolipidemic and antioxidant effects are attributed. In addition, they are found in the cell walls of plants, which defend them against ultraviolet radiation and pathogen attack [20], and can be used as an alternative for the treatment of breast, colon, and prostate cancer [6,21] as they inhibit tyrosinase; it is also mentioned that they can be used to inhibit the browning of fruits and vegetables [22].

Flavones are another group of compounds found in Mexican mistletoes [6]. These phytochemicals have been studied and widely used in different areas, and are part of the antioxidants involved in oxidation processes by eliminating free radicals [23]. In mistletoes of the Asian genus, flavones have been found with the capacity to reduce the presence of microorganisms such as fungi and bacteria, in addition to reducing blood pressure [24]. In *V. album* from Romania, hyperoside, isoquercitrin, rutin, and luteolin were found [17].

Anthocyanins, which belong to the subclass of flavonoids, are pigmented bioactive compounds [25]. Blackberries, raspberries, blueberries, black currants, strawberries, grapes, and spinach contain these compounds, which have the capacity to protect against a number of human diseases, in addition to having beneficial effects such as antioxidant, anti-allergic, antimicrobial, anti-inflammatory, antihyperglycemic, and anticarcinogenic properties [26,27,28].

For this study, four phytopathogens were isolated: *Alternaria alternata*, *Fusariun oxysporum*, *Fusarium* sp., and *Rhizoctonia solani* from a tomato crop; these fungi are the ones that cause the most losses in the crop. The characteristics observed under the microscope of *A. alternata* coincided with those described by Ronnie and Martinez [29], who mentioned that it is characterized by septate hyphae, ovoid to oblong, and clearly septate transversely and longitudinally. The coloration that was presented by *F. oxysporum* and the size of the macroconidia coincided with the description made by Espinoza-Ahumada et al. [30]. The observed macroscopic characteristics of *Fusarium* sp. are similar to those described by Lesly and Sumerell [31]. Ajayi and Bradley [32] mentioned that *R. solani* presented hyaline multinucleate septate hyphae, but with the passage of time, it changed to a brown color; in the distal dolichophore, the hypha was formed with a characteristic constriction at the branching point, and no conidia were observed.

The results obtained show that the *Phoradendron* sp. extract has the capacity to produce active substances (polyphenols) that inhibit the growth of *Alternaria alternata*. This extract has not been studied to determine its ability to suppress the growth of phytopathogens. Other extracts with fungistatic activity against this phytopathogen have been described. In this context, the crude alcoholic extract (100%) of *Thevetia peruviana* leaf against *Alternaria solani* showed 71.48% inhibition, while for the aqueous extract, 56.60% inhibition was observed at concentrations of 10 mg*L^−1^ (10000 ppm) [33]. Similarly, the maceration-acetone extract of *S. scandens* against *Alternaria solani* inhibited mycelial growth (92%) at 5000 ppm [34]. Meena et al. [33] mentioned that the effect of the *Thevetia peruviana* extract against *Alternaria solani* was through enzymatic reactions that caused the fungus to reduce the size of its conidia and increase the width of the mycelium. In addition, the cell wall was affected, and finally, mycelial death occurred.

Studies conducted by Tucuch et al. [35] indicated that ethanolic extracts of *Agave lechuguilla*, *Carya illinoinensis*, and *Lippia graveolens* inhibited 100% of the development of *Fusarium oxysporum*, from 250 mg*L^−1^ to 1000 mg*L^−1^; in this research, the antifungal activity barely exceeded 50% inhibition with the three extracts. There are studies with aqueous extracts of garlic, where it is reported that a 95% control of fungicidal activity against *F. oxysporum* caused the inhibition of lipid, nucleic acid, and protein synthesis [36]. It is important to note that there is research on the effect of extracts as oxidative stressors that elevate fungal toxins, and extracts of asparagus, corn, peas, and garlic can reduce the amount of fumonisin produced by the genus *Fusarium*, which is responsible for altering the plasma membrane. In plant and animal species [37], they disrupt biosynthesis, signaling and cellular functions, alter apoptosis and replication [38], and it is also reported that pineapple extracts increase the production of fumosinins, which is a disadvantage because it increases the toxicity of the fungus [39]. Studies conducted by Ochoa et al. [40] with cinnamon extract for the control of *Fusarium solani* showed different results to those shown in this research, since at 300 ppm, it showed an inhibition of mycelial growth of 45.58%, while the *Annona cheerimola* extract at a concentration of 4000 ppm inhibited 57.87%, results similar to those in this research, where at the same concentration, approximate values were obtained (44.91%) with the *Phoradendron* sp. extract grown on mesquite. Vásquez et al. [41] evaluated the inhibitory effect of aqueous extracts and essential oils of the *Chenopodium genus* on *F*. *solani*, and determined that the extracts of *Chenopodium ambrosioides* at 2% and *Chenopodium album* at 0.03% completely inhibited the growth of *F*. *solani* for up to 11 days, while the opposite occurred in this research, where the percentage of inhibition at the highest concentration reached 44.91%, for up to 7 days. Rodríguez et al. [42] mentioned that the extract of *Larrea tridentata* inhibited 100% the mycelial growth of *F*. *solani* for 10 days, while the extracts of *R. officinalis* and *B. squarrosa* also inhibited, but to a lesser degree. Research carried out for the control of *R. solani* with the methanolic extract of *L. tridentata* showed the results of inhibition of up to 100% of mycelial growth, the opposite of the case with the *Phoradendron* sp. extract, which had no significant inhibition for *R. solani*. As a result of this, it was concluded that the creosote bush extract had a better effect for the control of this pathogen [43]. Studies carried out by García et al. [6] with extracts of Mexican mistletoe *Phoradendron bollanum* and *Viscum album* subs. *austriacum* against phytopathogens such as *Xantomonas campestris, Clavibacter michiganensis, Alternaria alternata*, and *Fusarium oxysporum* determined that the saponins present in the extract were also used by the plants as a defense mechanism against pathogens, especially fungi. The results obtained by these authors were different to those in this study, since the *P*. *bollanum* extract inhibited 100% of the mycelial growth of *Alternaria alternata* from 200 mg*L^−1^, while for *Fusarium oxysporum* from 200 mg*L^−1^, it reached an inhibition of 70%, and for 1000 mg*L^−1^, it reached 80%, values higher than those observed in this research. The *Viscum album* subs. *austriacum* extract showed 50% of inhibition for *Alternaria alternata* at 200 mg*L^−1^, the inhibition increased to 70% at 1000 mg*L^−1^, and this extract presented double the percentage of inhibition (50%) at the same concentration (200 mg*L^−1^) compared to this study (22.46%). This extract had an inhibition of 50% of *Fusarium oxysporum* grow when 200 mg*L^−1^ was used and remained so up to 1000 mg*L^−1^; these results agree with those shown in this research where the extract of *Phoradendron* sp. grown on mesquite at 400 mg*L^−1^ had an inhibition of 48.88%.

Ramírez et al. [43] evaluated the production of conidia of *Alternaria solani* with clove extract, with which this fungus did not present conidia formation, and with the pepper extract, where the amount of conidia produced was higher than those obtained in this research. The extracts of *Phoradendron* sp. tested in this research were not able to totally inhibit the production of conidia of *F. oxysporum*, as in the case of the evaluation carried out by Ramírez et al. [43], where the clove extract obtained both by distillation and microwave and the cinnamon extract obtained by microwave were evaluated on the number of conidia formed by *F. oxysporum*, these extracts showed a total inhibitory effect. They also evaluated microwave derived cinnamon, clove, and pepper oils with the same pathogen, which also did not allow for conidia formation. Finally, in research conducted with *Fusarium* sp. oils by Vásquez et al. [41], where they evaluated essential oils (0.3–2%) of *Chenopodium album* and *Chenopodium ambrosioides*, it showed an antifungal effect on *Fusarium solani* and *F. oxysporum*, since they caused a significant reduction in mycelial growth and conidia production.

## 4. Materials and Methods

### 4.1. Obtaining Plant Material

Fresh leaves and stems of *Phoradendron* sp. were collected from three different hosts: from mesquite was collected in Cuatrociénegas, and from oak and cedar in Arteaga, both counties being located in the State of Coahuila, Mexico. The collected plant material was placed in an oven to dry for 72 h at 60 °C, and then ground to a fine powder.

#### 4.1.1. Ultrasound-Microwave Assisted Extraction

To obtain the extracts, 62.5 g. of the powder samples were mixed in 1000 mL of distilled water, and placed in the ultrasound and microwave equipment (Nanjing ATPIO Instruments Manufacture Co. Ltd. Company, China) with the following conditions. Ultrasonic (VS): Power Radio 20, Ultrasonic on Relay 10, Ultrasonic off Relay 3, Amplitude off Relay 25, and Set Time 20. Microwave (MV): Power Radio 800, Display power 0, Set Temp 70 °C, and Holding Time 5. Once the extraction was completed, the samples were filtered through organza cloth and filter paper, recovering approximately 700 mL of each specie. The extracts were then stored in deep freezing at a temperature of −81 °C.

#### 4.1.2. Column Chromatography with Amberlite

Column chromatography was carried out using XAD-16N amberlite as the stationary phase, previously activated with methanol for 10 min and placed in the column. Subsequently, 300 mL of the extract was placed at the top of the column and distilled water was added as a mobile phase to remove water-soluble compounds and then ethanol was added for the recovery of polyphenols [44]. Once the three fractions had been obtained, they were distributed in glass containers and dried in an oven at 60 °C, without exposure to light for 24 to 48 h. Finally, the dry extract was collected in a powder form and stored in an amber flask at room temperature for further analysis.

#### 4.1.3. Characterization of Phytochemicals Present in the Plant Extracts Using RP-HPLC-ESI-MS Liquid Chromatography

The analysis by reversed-phase high performance liquid chromatography was performed following the methodology of Ascacio-Valdés et al. [45], which consists of using a Varian HPLC system including an automatic injector (Varian ProStar 410, USA), a ternary pump (Varian ProStar 2310, USA), and a PDA decanter (Varian ProStar 330, USA). A liquid chromatograph ion trap mass spectrometer (Varian 500-MS IT Mass Spectrometer, USA) equipped with an electrospray ion source was also used. Samples (5 μL) were injected into a Denali C18 column (150 mm × 2.1 mm, 3 μm, Grace, USA). The oven temperature was maintained at 30 °C. The eluents were formic acid (0.2%, *V*/*V*; solvent A) and acetonitrile (solvent B). The following gradient was applied: initial, 3% B; 0–5 min, 9% linear B; 5–15 min, 16% linear B; 15–45 min, 50% linear B. Then, the column was washed and reconditioned, the flow rate was maintained at 0.2 mL*min^−1^, and the elution was controlled at 245, 280, 320, and 550 nm. The entire effluent was injected (0.2 mL*min^−1^) into the mass spectrometer source without splitting.

### 4.2. Tomato Pathogen Isolation

Samples were collected in the municipality of Altamira, Tamaulipas, Mexico, from tomato plantations of saladette tomato variety 8579, where soil and tissue samples with symptoms of disease were taken. Root and stem tissues were cut into segments of approximately 5 mm in diameter, disinfected with a 3 (*v*/*v*), sodium hypochlorite solution for 1 min, then rinsed three times in sterile distilled water, and left to dry on sterile blotting paper under a laminar flow hood. Once dried, samples were placed in Petri dishes with potato dextrose agar (PDA) culture medium and incubated at 28 °C, with daily observations until mycelial growth. Phytopathogen isolation from the soil samples was carried out using the serial dilution technique; 10 g of soil was added to 90 mL of sterile distilled water, and agitated for 10 min in a shaker, then 1 mL of this solution was taken and placed in a tube containing 9 mL of sterile distilled water, agitated with the vortex, and then 1 mL was taken from this tube and passed to the next one until reaching a dilution of 1 × 10 ^−5^. From the 1 × 10 ^−4^ and 1 × 10 ^−5^ dilution tubes, 500 μL was taken and placed in Petri dishes containing PDA culture medium and subsequently incubated under the same conditions as described for the plant tissue samples.

Once mycelial growth was observed in the Petri dishes containing both samples, we proceeded to hyphal tip purification. Morphological identification was performed by microscopic observation of the reproductive structures and parts of the mycelium of each phytopathogen, using specialized taxonomic keys for identification [46].

#### Molecular Identification of Isolated Fungi

Total genomic DNA extraction and purification were performed using the commercial Wizard^®^ Genomic DNA Purification Kit following the manufacturer’s recommended plant DNA extraction protocol. The basic steps involved in DNA isolation are: (1) Disruption of the cell structure to create a lysate; (2) protection of DNA from degradation during processing; (3) separation of the soluble DNA from cell debris and other insoluble material; and (4) elution of purified DNA. Forty mg of surface mycelium scraped from fungi grown on PDA agar was used. The DNA obtained was quantified by spectrophotometry (260/280 nm) using the EPOCH Kit (Biotek). Integrity was verified by 1 (*w*/*v*) agarose gel electrophoresis. DNA was stored at −20 °C until amplification. The ITS region of the ribosomal DNA was amplified by PCR using the universal primers ITS2 (5′GCTGCGTTCTTCATCGATGC3′) and ITS5 (5′GGAAGTAAAAGTCGTAACAAGG3′) [47] at 10 µM Master Mix (Invitrogen) and 100 ng of DNA. The final PCR reaction volume was 25 μL, the thermal profile for this simple PCR was standardized with a denaturation temperature of 95 °C for five minutes, a 25-cycle phase at 94 °C for one minute, 50.3 °C for one minute, and 72 °C for one minute, and then a final extension phase at a temperature of 72 °C for five minutes. The PCR products obtained were visualized by 1 (*w*/*v*) agarose gel electrophoresis at 85 v, 400 mA for 30 min, stained with 2 μL of ethidium bromide and sent to the company Psomagen. Fungal identification to the genus and species level was performed using the Basic Local Alignment Search Tool (BLAST) of the National Center for Biotechnology Information (NCBI) [48].

### 4.3. Antifungal Activity Using Poisoned Medium

The antifungal activity was determined by means of the poisoned medium technique using sterilized potato dextrose agar (PDA) culture medium. A stock solution of 10,000 ppm of the different extracts was prepared to be mixed with the culture medium and to prepare the treatments in concentrations of 250, 500, 1000, 2000, and 4000 ppm; the mixture was poured into Petri dishes of 80 mm in diameter and remained for 24 h to verify its sterility. Subsequently, in the center of each Petri dish containing the extract, a disc of mycelium (diameter 5 mm) of the fungi grown on PDA was deposited; the boxes were sealed and incubated at 28 ± 2 °C. The diameter of fungal growth was measured every 24 h until the control strain without the extract application completely covered the Petri dish. The percentage inhibition was determined according to Equation (1).
% inhibition = (DC − DT/DC) × 100(1)
where DC is the fungal diameter with the control treatment, and DT is the fungal diameter with the different extract concentrations.

### 4.4. Number of Conidia

This parameter was evaluated by selecting a repetition of each fungus evaluated for each dose including the control, with each of the extracts, from which a mycelium disk of 5 mm was taken at a distance of 15 mm from the center of the Petri dish and placed in a tube with 1 mL of sterile distilled water. The sample was shaken in a vortex and placed using a micropipette in the Neubauer chamber, where the conidia count was performed taking the four squares at the ends and the one in the center, then with the data obtained, the effect that the extracts had on the number of conidia*mL produced by each fungus was determined using Equation (2) proposed by Ruiz et al. [49]
 C/mL = Number of conidia × 10000/Number of squares counted × Dilution factor (2)

### 4.5. Statistical Analysis

The experiment was established under a completely randomized design, with four replications for each treatment. Then, with the obtained results, an analysis of variance was performed. When needed, Tukey tests (*p* < 0.05) were employed for treatment means comparison.

## 5. Conclusions

The phytochemical compounds contained in *Phoradendron* sp. have fungistatic activity on *Alternaria alternata*, *Fusarium* sp., and *F. oxysporum*, however, they have a limited effect against *Rhizoctonia solani*. This study shows that extracts of *Phoradendron* sp. growing on oak, mesquite, and cedar are promising for the control of the main phytopathogenic fungi of the tomato crop in southern Tamaulipas, Mexico, and we recommend continuing studies into the polyphenols of these extracts with other phytopathogens of agricultural importance. After the studies in laboratory conditions, and with the results presented in this research, it is recommended that tests are carried out in crops under open field conditions. Since these extracts are elaborated from hemiparasite plants that are considered pests in Mexico, there will be no deforestation of species or damage to the environment.

## Figures and Tables

**Figure 1 plants-12-00672-f001:**
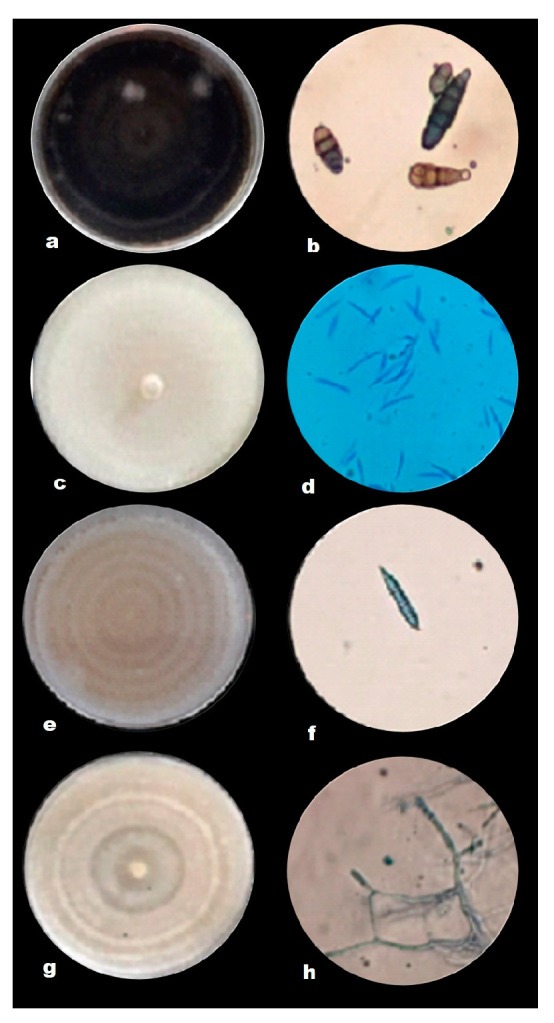
(**a**) *Alternaria alternata* in Petri dish; (**b**) *Alternaria alternata* conidia under microscope; (**c**) *Fusarium* sp. in Petri dish; (**d**) *Fusarium* sp. conidia; (**e**) *Fusarium oxysporum* in Petri dish; (**f**) *Fusarium oxysporum* viewed under microscope; (**g**) *Rhizoctonia solani* in Petri dish; and (**h**) *Rhizoctonia solani* observed under a microscope.

**Table 1 plants-12-00672-t001:** Composition of the *Phoradendron* sp. extract on different hosts.

Extract	T.R (min)	Mass(m/z)	Compound (70% 1:12)	Family
CME	15.83	353.1	1-Caffeoylquinic acid	Hydroxycinnamic acids
	18.03	353.0	3-Caffeoylquinic acid	Hydroxycinnamic acids
	21.36	353.0	4-Caffeoylquinic acid	Hydroxycinnamic acids
	35.85	325.1	p-Coumaric acid 4-O-glucoside	Hydroxycinnamic acids
	38.49	597.1	Delphinidin 3-O-sambubioside	Anthocyanins
	39.65	597.1	Delphinidin 3-O-sambubioside	Anthocyanins
	40.53	597.1	Delphinidin 3-O-sambubioside	Anthocyanins
	30.37	381.1	Quercetin 3′-sulfate	Flavonols
OME	15.79	353.0	1-Caffeoylquinic acid	Hydroxycinnamic acids
	18.74	353.1	3-Caffeoylquinic acid	Hydroxycinnamic acids
	34.41	285.1	Luteolin	Flavones
	40.13	285.0	Kaempferol	Flavonols
MME	15.40	353.0	1-Caffeoylquinic acid	Hydroxycinnamic acids
	18.21	353.0	3-Caffeoylquinic acid	Hydroxycinnamic acids
	33.36	285.1	Luteolin	Flavones
	39.17	285.0	Kaempferol	Flavonols
	40.59	285.0	Scutellarein	Flavones

CME = mistletoe grown on cedar, OME = mistletoe grown on oak, MME = mistletoe grown on mesquite.

**Table 2 plants-12-00672-t002:** BLAST homolog search results in NCBI.

Scientific Name	Query Cover	Per. Ident	Accession
*Alternaria alternata*	99	99.55	MT446176.1
*Fusarium oxysporum*	49	98.99	MF630984.1
*Fusarium* sp.	100	100	MH884139.1
*Rhizoctonia solani*	92	99.53	KX674533.1

**Table 3 plants-12-00672-t003:** Percent inhibition of mycelial growth of tomato phytopathogenic fungi when grown on a medium poisoned by polyphenols obtained from the mistletoe extracts of different hosts.

Extract	Concentration (ppm)	*A. alternata %*		*F. oxysporum %*		*Fusarium* sp. *%*		*R. solani %*	
	Control	0	d	0	c	0	d	0	b
	250	7.65	c	12.72	b	19.12	b	0	b
CME	500	23.63	ab	13.03	b	18.55	b	0	b
	1000	19.76	bc	19.04	ab	15.83	c	0	b
	2000	22.46	abc	24.30	ab	15.36	c	0	b
	4000	35.60	a	13.13	b	25.99	a	1.70	a
	Control	0	c	0	e	0	e	0	b
	250	2.52	c	18.11	d	17.36	b	0	b
OME	500	13.72	ab	17.87	d	12.38	d	0	b
	1000	11.65	bc	30.88	c	14.15	c	0	b
	2000	10.56	bc	37.23	b	23.61	a	0	b
	4000	24.00	a	40.60	a	24.16	a	1.33	a
	Control	0	d	0	e	0	d	0	c
	250	13.50	bc	17.68	d	3.50	c	0	c
MME	500	16.85	b	25.36	c	3.82	c	0	c
	1000	12.86	c	40.91	b	3.94	c	0	c
	2000	13.84	bc	49.36	a	11.66	b	3.47	b
	4000	22.03	a	52.32	a	44 75	a	13.05	a

Means with equal letters between columns are statistically equal (*p* < 0.05).

**Table 4 plants-12-00672-t004:** Effect of the plant extracts of mistletoe growth on different hosts on the number of conidia/mL produced by *Alternaria alternata*, *Fusarium oxysporum*, and *Fusarium* sp.

		Concentrations (ppm)											
Fungus	Extract	Control		250		500		1000		2000		4000	
*Alternaria* *alternata* Conidia (1 × 10^6^ mL^−1^)	CME	25.2	a	3.6	b	3.8	b	0.8	b	2.6	b	0.2	b
	OME	25.2	a	2.4	c	0.2	c	3.6	bc	2.4	c	6.8	b
	MME	25.2	a	0.8	b	3.6	b	0.4	b	3.4	b	5.6	b
*Fusarium* *oxysporum* Conidia (1 × 10^6^ mL^−1^)	CME	549.2	a	167.4	de	215.8	cd	354	b	542.8	a	249.4	c
	OME	272.2	a	117.4	bc	69.4	d	126.4	b	84.8	cd	117.4	bc
	MME	121.4	a	59.8	b	42	b	59.8	b	58	b	55.8	b
*Fusarium* sp. Conidia (1 × 10^6^ mL^−1^)	CME	30	a	5.8	b	5.8	b	9.2	b	4.6	b	14.8	b
	OME	42.8	b	3.8	de	15	c	9.2	cd	11.4	cd	53	a
	MME	37.8	b	55.4	ab	84.6	a	62.6	ab	20.8	b	34	b

CME = cedar mistletoe extract, OME = oak mistletoe extract, MME = mesquite mistletoe extract. Note: Means of inhibition between doses with the same letter are statistically equal.

## Data Availability

Not applicable.

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
