# Peer review of "Antifungal In Vitro Activity of Phoradendron sp. Extracts on Fungal Isolates from Tomato Crop"

_plants, 2023, doi:10.3390/plants12030672_

Round 1

Reviewer 1 Report

The manuscript is interesting and the presented data is new and of practical merit, but the text requires significant improvements before it can be recommended for publication, as listed below and in the corrected manuscript.

This is a quite short and preliminary study, therefore, it should be categorized as communication instead of an article.

Keywords should be arranged alphabetically.

The text is understandable, but the English form (style, grammar, and punctuation) requires significant improvements. Some sentences are very long and should be divided into shorter ones. MDPI uses a serial comma.

Some parts of the Results are in fact a Discussion (see the corrected manuscript).

All abbreviations in Tables must be explained in the footnote. Units in Table 1 are missing. What was the concentration of the compounds presented in Table 1?

Figure 1 lacks scale bars.

Standard errors are missing in Table 4.

Avoid repetitions in the text.

The unit style requires correction.

Some parts of the Discussion are unclear.

Please name exactly what species of mistletoe and cultivar of tomato were studied.

More detailed information on the producers of key chemicals and equipment is needed.

Additional information in the Materials and methods chapter is needed, e.g. on the molecular analyses. Some parts of this chapter are unclear to me.

For more specific comments, please see the corrected manuscript.

Reviewer 2 Report

Review report

Comments and suggestions to the authors

A brief summary

The manuscript is based on the obtaining of polyphenol extracts from mistletoe plants growing on three different tree species - mesquite (Prosopis glandulosa), cedar (Cedrus) and oak (Quercus). The overall chemical structure of the obtained plant extracts was investigated by RP-HPLC-ESI-MS liquid chromatography. The main goal of this investigation is an alternative biocontrol strategy for management of plant diseases caused by fungi to be studied. The antifungal effect of these extracts was screened. The target phytopathogenic fungi have been isolated from tomato plantation located in Altamira, Tamaulipas, Mexico. The microorganisms were characterized by classical and molecular methods and identified as Alternaria alternata, Fusarium oxysporum, Fusarium sp. and Rhizoctonia solani.

My opinion is that studies reporting new and effective strategies in combating plant diseases are very necessary and should be supported. However, I have some remarks and suggestions:

Title: The Latin name of mistletoe (Phoradendron sp.) is not mentioned in the main text as the main plant source of extraction of polyphenols. Instead, the commercial name "mistletoe" is used in whole manuscript with few exceptions. My suggestion is either change the name in the title with mistletoe or in the text with its species name. In scientific manuscript is better to use the scientific name of the plant species or to give the commercial one in brackets after the first appearance in the text.

Abstract:

Line 23: separate different concentrations with commas.

Introduction:

Line 56-59: I suggest here to be more specific. You should give information concerning the exact phytochemical compounds in order to become clearer what kind of secondary metabolites have been extracted from the different mistletoe species. Moreover, specific information about their antimicrobial action is also missing in the text.

Materials and Methods:

General remark: sections 4.2., 4.3. and 4.4. could be combined in one general Section 4 with three subsections because all the information there is about the initial processing of the plant extracts.

Line 257: “samples” should be in plural as the samples are three, not one;

Line 272: How many fractions have you obtained? One or three? You should be more specific here. You did extractions from three specimens so the fractions should be three. Am I right?

Line 290:” was injected” is a substitution and should be erased;

Line 293: You wrote: “Sample from soil and tissue with disease symptoms….” From one tomato plant or from more than one. How many plant samples have been screened for presence of pathogenic fungi? You should add this information.

Line 297-299: “Once dried, samples were placed in Petri dishes with 297 Potato-Dextrose-Agar (PDA) culture medium and incubated at 28 °C, with daily observations until mycelial growth.” Without preparing a solution? Please clarify here.

Line 301: What media did you used? Add this information.

Line 315: ITS2_ITS5 – you should add reference for the primers.

Line 317: PCR products: the PCR conditions should be added. The expected length of the amplicon should be added too;

Line 319: I suppose that the amplification products have been purified and sequenced. Add this information here.

Line 325: specify the culture medium;

Line 326-327: “mistletoe polyphenol treatments….” did you sterilize the mistletoes solutions prior adding them to the culture media? Please, add this important information. Specify the method of solution sterilization.

Line 327: It is not correct to write 0 concentration. Remove this from the text and specify in separate sentence that these are the control petri dishes - without extracts.

Results: This section represents in logical order all obtained results.

Table 1: give full names of CME, OME, MME and T.R. as these are the first place in the text where these abbreviations are mentioned.

Line 77: I suggest here to substitute the word “phytopathogens” with “phytopathogenic fungal isolates”. This is just a suggestion.

Line 77: Please clarify the exact origin of each isolate.

Line 80: “with the passage of days…” during the cultivation period? Not specific here...

Line 85 and 87: “50 μm in length” - haw did you measured the length of the conidia? There are no bars in the Figure 1.

Line 86: “5 septa” - not visible in Figure 1;

Line 130: for OME - the percent inhibition at 4000 ppm given in the Table 3 does not correspond to those written here. Please, clarify which value is the correct one.

Line 138: 106… - ten to the sixth degree I suppose...

Discussion

General rematks: Many sentences are far too long and unclear. You should pay attention on the Latin name of the species. In many places they are given with their full names or abbreviated but have to be vice versa (lines: 228, 230, 231, 234, 147 and so on…)

Line 156: “have antioxidant…” You should write “have been reported to have...” The antioxidant and antibacterial effects of yours compounds have not been presented in this study.

Line 158-161: This sentence is too long and not clear. You should separate it in order to be clearer.

Line 168-177: this passage is not clear and too long. Separate it into 3-4 sentences in order the text to become clearer.

Line 214-219: this passage is not clear and too long. Separate it into 3-4 sentences in order the text to become clearer.

Line 219-226: this passage is not clear and too long. Separate it into 3-4 sentences in order the text to become clearer.

Line 239 – 246: It is absolutely hard for me to read this part. Each sentence is far too long. You should optimize the text.

Reviewer 3 Report

This article presented Antifungal in vitro activity of Phoradendron sp. extracts on fungal isolates from tomato crop. This study will facilitate assessment of tomato pathogens and potential of plant extracts against these pathogens. Before recommending this article for publication, there are some shortcomings for that should be resolve.

Add commas here “250 500 1000 2000 and 4000 ppm and a control”

The abstract should briefly describe sample collection methods and details of extract preparations.

In introduction add significance and importance of the plants used for extract preparation.

Also add its antimicrobial potential.

Line 47 should be cited with recent study. The following study could be helpful.

https://doi.org/10.1016/j.pmpp.2021.101639,

To determine potential of plant extracts is in vitro study is enough? Please justify.

Figure 1 morphology of the strains should be clearly presented.

Section 4.7 should be cited with relevant study.

https://doi.org/10.1016/j.bcab.2020.101729,

In discussion the authors should present that which phytochemicals may be responsible for antifungal activity based on the previous or relevant literature.

Also modify conclusion by adding future recommendations.

Lack of in vivo study is the limitation of this study.

Round 2

Reviewer 1 Report

The authors responded to all querries. The manuscript is now suitable for publication.